

# How does soil water availability control phytotoxic O₃ dose to montane pines? Modelling and experimental study from two contrasting climatic regions in Europe

Svetlana Bičárová[1], Zuzana Sitková[2], Hana Pavlendová[2], Peter Fleischer jr.[3], Peter Fleischer sr.[3], Laurence Dalstein-Richier[4], Marie-Lyne Ciriani[4] and Andrzej Bytnerowicz[5]

[1]Institute of Earth Science of the Slovak Academy of Sciences, Stará Lesná, 059 60 Tatranská Lomnica, Slovakia
[2]National Forest Centre–Forest Research Institute Zvolen, T. G. Masaryka 22, 960 92 Zvolen, Slovakia
[3]Technical University in Zvolen, T. G. Masaryka 24, 960 92 Zvolen, Slovakia
[4]Groupe International d'Etudes des Forêts Sud-européennes G.I.E.F.S, 69, Avenue des Hespérides, 06300 Nice, France
[5]USDA Forest Service, Pacific Southwest Research Station, 4955 Canyon Crest Drive, Riverside, CA 92507, USA

*Correspondence to*: Svetlana Bičárová (bicarova@ta3.sk)

**Abstract.** Montane forests are exposed to high ambient ozone (O₃) concentrations that may adversely affect physiological processes in internal cells when O₃ molecules enter the plants through the stomata. This study addresses the model results of Phytotoxic Ozone Dose metric (POD) based on estimation of stomatal O₃ flux to dwarf mountain pine (*Pinus mugo)* and Swiss stone pine *(Pinus cembra)*. We focused on two different bioclimatic regions: (1) the temperate mountain forests in the High Tatra Mts (SK–HT) of the Western Carpathians, and (2) the Mediterranean forests of the Alpes–Mercantour (FR–Alp) in the Alpes–Maritimes. Field measurement of O₃ concentration and meteorological data incorporated into deposition model DO₃SE showed lower O₃ flux in FR–Alp than in SK–HT plots for the 2016 growing season. Model outputs showed that soil humidity play a key role in stomatal O₃ uptake by montane pines at the alpine timberline. We found that temperate climatic conditions in SK–HT with sufficient precipitation did not limit stomatal conductivity and O₃ uptake of *P. mugo* and *P. cembra*. On the other hand, the Mediterranean mountain climate characterised by warm and dry summer reduced stomatal conductance of pines in FR–Alp. POD without threshold limitation i.e. POD₀ as a recently developed biologically sounded O₃ metric varied near around and below critical level (CLef) depending upon different conditions of sunshine exposure in SK–HT plots. Field observation at these plots showed relatively weak visible O₃ injury on *P. cembra* (2 % and 7 %) when compared with *P. mugo* (8 % and 18 %) for one year (C+1) and two year (C+2) old needles, respectively. Despite of low POD₀ values, clearly below CLef, the highest level of visible O₃ damage on average from 10 % (C+1) to 25 % (C+2) was observed on *P. cembra* needles in Mediterranean (FR–Alp) area. Further research is needed to clarify the effect of real soil moisture regime on stomatal closure in dry areas (FR–Alp) and resistance of pine species against visible O₃ injury in wet subalpine zones (SK–HT). More attention should be paid to O₃ fluxes covering a year-round growing season as well as intra-daily dynamics, especially the night hours, since these time spans appear to play significant role in O₃ uptake by mountain conifers.



# 1 Introduction

Surface ozone ($O_3$) is one of the most common air pollutants according to on a recent review of the accumulated scientific evidence (WHO, 2006; EPA, 2014; EEA, 2016; UNECE, 2016). During recent decades, trends in mean $O_3$ concentrations have varied by regions (Cooper et al., 2014) nevertheless did not appear to be well associated with some exposure metrics applicable for assessing human health or vegetation effects (Lefohn et al., 2017). A variety of $O_3$ metrics are used in the risk assessment for forest trees. Initially developed (or original) exposure standards (AOT40) based only on measured $O_3$ concentration does not take into account environmental factors affecting responses of vegetation. Therefore in late 1990s a discussion on developing new, flux-based critical levels, started. New approaches focus on the principles of $O_3$ transport from atmosphere to the plant interior through the stomata, and control of $O_3$ uptake by leaves *via* environmental factors (Fuhrer et al., 1997; Massman et al., 2000; Grünhage et al., 2001; Ashmore et al., 2004; Musselman et al., 2006; Karlsson et al., 2007; Matyssek et al., 2007). A stomatal conductance based model was developed to estimate $O_3$ uptake for a number of the most widespread tree species (Emberson et al., 2000). The new flux-based critical levels revised by the LRTAP Convention (CLRTAP, 2015; Mills et al., 2011) were named as the Phytotoxic Ozone Dose ($POD_Y$), i.e., the accumulated stomatal $O_3$ flux above a threshold (Y) flux. There is strong support among biologists for the use of the threshold $O_3$ flux that includes the detoxification capacity of the trees (Karlsson et al., 2007). Expert judgement was used to set Y=1 nmol $m^{-2}$ PLA $s^{-1}$ (PLA is the projected leaf area) based on observation of $O_3$ sensitivity under controlled conditions (Dizengremel et al., 2013). De Marco et al. (2015) recommend applying $POD_Y$ without threshold limitation (Y=0) i.e., $POD_0$ rather than $POD_1$. It is based on the fact that any $O_3$ molecule entering into the leaf may induce a metabolic response (Musselman et al., 2006). Various studies have provided information on how $O_3$ interacts with the plant at the cellular level (Bussotti et al., 2011; Gottardini et al., 2014; Braun et al., 2014; Mills et al., 2016). In addition, the physiological consequences of the $O_3$ induced effects may impair resistance of trees to the abiotic (frost, drought) and biotic (nutrient deficiencies, pathogens, bark beetle) stress factors (Vollenweider and Günthardt-Goerg, 2006).

The major challenge in the development of $O_3$ standards is their validation against biologically based field data (Paoletti and Manning, 2007). Recent epidemiological studies show better correlation between $POD_0$ and visible foliar $O_3$ injury than AOT40 (Sicard et al., 2016). The most sensitive conifers are *Pinus* species (Dalstein and Vas, 2005), however different visible $O_3$ injury response may be expected under natural conditions (Coulston et al., 2003; Nunn et al., 2007; Braun et al., 2014). Based on large literature evidence, mountain forest in the Carpathians (Bytnerowicz et al., 2004; Hůnová et al., 2010; Zapletal et al., 2012; Bičárová et al., 2016), and in the Alps (Smidt and Herman, 2004; Sicard et al., 2011) are exposed to high $O_3$ concentrations. Ozone damage rates increase with altitude in response to increasing $O_3$ mixing ratios and $O_3$ uptake due to favourable microclimatic conditions (Díaz-de-Quijano et al., 2009). However there is still lack of empirical data concerning vulnerable mountain forest tree species.

The objectives of this study were: (i) to map ozone metrics (AOT40, $POD_1$, $POD_0$) for the growing season 2016 and assess them with respect to related critical levels including the innovative species specific flux-based critical level (CLef) for forest





protection against visible $O_3$ injury; (ii) to appraise the role of physiological parameters and soil water availability in $O_3$ uptakes
under contrasting climate conditions; and (iii) to analyse the relation between model results of $POD_0$ and field observation of visual $O_3$ injury for high $O_3$ sensitive conifers such as Swiss stone pine (*Pinus cembra* L.) and dwarf mountain pine (*Pinus mugo* Turra). To achieve these goals we focused on two different mountain bioclimatic regions of Europe such as (1) the High Tatra Mts (SK–HT) in the Western Carpathians with temperate climate and (2) the Alpes-Mercantour (FR–Alp) in the Alpes-Maritimes where forest vegetation is influenced by Mediterranean climate.

**2 Study area**

The study area (Fig. 1) covers montane forest sites situated in the Tatra National park (SK–HT) and in the Mercantour National Park (FR–Alp). SK–HT region is the highest mountain range of the Western Carpathians located in the north Slovakia. The elevation in this region rises from foothills at 800 m a.s.l. to the highest peak at 2,655 m a.s.l.. Climate is mostly cold and humid. According to standard reference climate period (1961–1990), mean annual air temperature ranges from 5.3 °C at foothills
to –3.8 °C in zone above 2,600 m a.s.l. Mean annual precipitation ranges from 760 to 2,000 mm. In the growing season (Apr–Sep) precipitation reaches nearly 65 % of annual sum with culmination in June or July. The coniferous forests cover area up to 1,600 m a.s.l. Subalpine zone (up to 1,800 m a.s.l.) is almost completely covered by dwarf mountain pine. Swiss stone pine occurs sparsely at the timberline. In recent decades, the massive windstorms and consecutive bark beetle (*Ips typographus)* outbreaks damaged SK–HT forests weakened by various abiotic factors such as elevated temperature (Mezei et al., 2017) or
long-range transport of air pollutants (Bytnerowicz et al., 2004).
The Mercantour National Park with several peaks above 3,000 m, and wide altitudinal range of more than 2,500 metres has a specific climate mixing Mediterranean and Alpine influences. In conjunction with multiform geological terrains ranging from limestone to crystalline, the habitats of the southwestern Alps are extremely diverse and this region belongs to the richest hotspots of biodiversity in Europe (Dole-Olivier et al., 2015). Climate in (FR–Alp) is temperate Mediterranean with moderately
cool and dry summer. Minimum monthly precipitation occurs in July, exactly the opposite as in SK–HT region. In this region the Holm oak, the Mediterranean olive tree, rhododendron, fir, spruce, Swiss stone pine and larch are widespread species. Previous studies have shown (Dalstein and Vas, 2005; Sicard et al., 2011) that the rural alpine Mediterranean area of the Mercantour National Park may be affected by considerable quantities of localized $O_3$ generated by photochemical reactions of $O_3$ precursors from the regional road traffic in presence of high temperatures and solar radiation in very hot summers of the
Mediterranean climate. Visible leaf injury on particularly sensitive species is one of the $O_3$ air pollution symptoms.
Six study sites were established in different altitudinal zones to map exposure and phytotoxic $O_3$ doses in contrasting climate conditions: A–foothills, B–submontane zone, C–subalpine zone, D–subnival zone and different aspects: 1 – south, 2 – north (Table 1, Fig. 1).




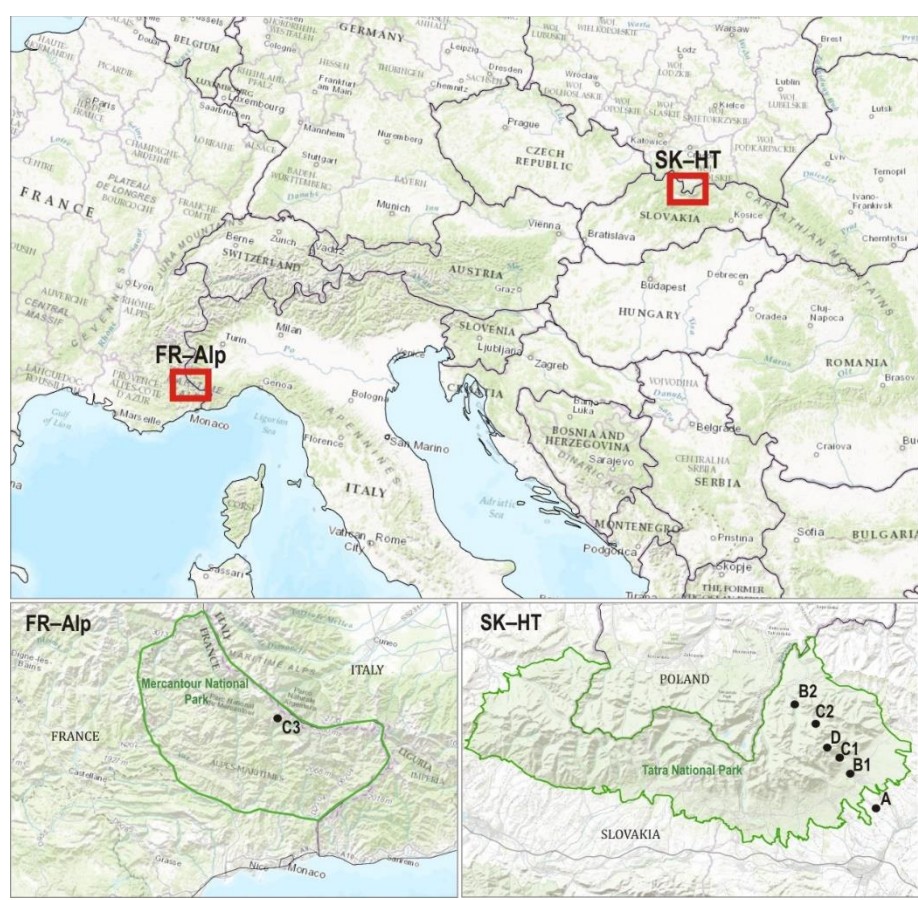

Figure 1. The geographical position of the Tatra National Park (SK–HT) in the Carpathian mountain range and the Mercantour National Park in the Alpes-Maritimes region (FR–Alp) including experimental sites in order of altitudinal zonation; A–foothills: Stará Lesná; B–submontane zone: B1–Štart, B2–Podmuráň; C–subalpine zone: C1–Skalnaté pleso, C2–Kolové pleso, C3–Col de Salèse; D–subnival zone: Lomnický štít.

Table 1. Description of experimental sites (CODE used in relation to Fig. 1)

| CODE site | GPS Latitude Longitude | Altitude Zone/Aspect | Climate (Köppen classification) Local specification | Tree species composition | Soil type Soil texture |
|---|---|---|---|---|---|
| SK–HT | | | | | |
| A Stará Lesná | 49°09'08" N 20°17'19" E | 810 m a.s.l. Foothill/S | Temperate continental (Dfb); Moderately warm and humid | *Picea abies, Betula verucosa, Pinus sylvestris, Larix decidua, Alnus glutinosa* | Haplic Cambisols; Silt loam (medium coarse) |
| B1 Štart | 49°10'30"N 20°14'48" E | 1,150 m a.s.l. Submontane/S | Temperate continental (Dfb); Moderately cool and humid | *Picea abies, Larix decidua, Pinus sylvestris, Abies alba, Acer pseudoplatanus* | Haplic Podzols; Sandy loam (coarse) |
| B2 Podmuráň | 49°15'00" N 20°09'25" E | 1,100 m a.s.l. Submontane/N | Temperate continental (Dfb); Moderately cool and extremely humid | *Picea abies, Abies alba, Sorbus aucuparia, Fagus silvatica, Acer pseudoplatanus* | Haplic Podzols; Loam (medium) |
| C1 Skalnaté pleso | 49°11'21" N 20°14'02" E | 1,778 m a.s.l. Subalpine/S | Cool continental (Dfc); Cool mountain and extremely humid | *Pinus mugo* | Folic Leptosols; Sandy loam (coarse) |



| | | | | | | |
|---|---|---|---|---|---|---|
| C2 Kolové pleso | 49°13'22" N 20°11'27" E | 1,570 m a.s.l. Subalpine/N | Cool continental (Dfc); Cool mountain and extremely humid | *Pinus mugo, Pinus cembra, Picea abies* | | Folic Leptosols; Silt loam (medium coarse) |
| D Lomnický štít | 49°11'43" N 20°12'54" E | 2,635 m a.s.l. Subnival summit | Cool continental (Dfc); Cold mountain and extremely humid | : | | Lithic Leptosols; Hyperskeletic |
| FR–Alp | | | | | | |
| C3 Col de Salèse | 44°08'18" N 07°14'11" E | 1,993 m a.s.l. Subalpine/S | Temperate Mediterranean (Csb); Moderately cool and dry summer | *Pinus cembra, Larix decidua* | | Lithosols Hyperskeletic |

## 3 Methods

### 3.1. Ozone metrics

All $O_3$ metrics (AOT40, $POD_1$, $POD_0$) were calculated using the multiplicative model $DO_3SE$ (Deposition of Ozone for Stomatal Exchange). AOT40 (ppb h) is the accumulated amount of ozone over the threshold value of 40 ppb for daylight hours
during the relevant growing season (Apr–Sept). Concentration based critical level CLec of AOT40 was set to 5,000 ppb h (Directive 2008/50/EC). An algorithm for model estimation of $POD_Y$ (mmol m$^{-2}$ PLA) incorporates effects of meteorological conditions such as air temperature ($f_{temp}$), vapour pressure deficit ($f_{VPD}$), solar radiation or light ($f_{light}$), furthermore soil water potential ($f_{SWP}$), plant phenology ($f_{phen}$) and $O_3$ concentration ($f_{O3}$) on the maximum stomatal conductance ($G_{max}$). Passage rate of $O_3$ entering through the stomata is expressed as the stomatal $O_3$ conductance $G_{sto}$ (m s$^{-1}$):

$$G_{sto} = G_{max} * [min(f_{phen}, f_{O_3})] * f_{light} * max\{f_{min}, (f_{temp} * f_{VPD} * f_{SWP})\} \qquad (1)$$

Stomatal $O_3$ flux $F_{st}$ (nmol m$^{-2}$ PLA s$^{-1}$) is then given by:

$$F_{st} = G_{sto} * c(z_1) * \left(\frac{r_c}{(r_b+r_c)}\right) \qquad (2)$$

where $c(z_1)$ is concentration of $O_3$ (nmol m$^{-3}$) at the top of the canopy measured in the tree height ($z_1$), $r_b$ and $r_c$ are the quasi laminar resistance and the leaf surface resistance (s m$^{-1}$), respectively.

Phytotoxic ozone dose $POD_Y$ is sum of hourly values of $F_{st}$ (Eq. 2) over threshold Y=1 ($POD_1$) or without threshold Y=0 ($POD_0$) aggregated over the growing season. Stomatal flux-based critical levels $CLef_1$ of $POD_1$ is proposed to be 8 mmol m$^{-2}$ PLA for evergreen coniferous, especially Norway spruce (Mills et al., 2011). An innovative species-specific CLef of $POD_0$ is proposed to be 19 mmol m$^{-2}$ PLA for forest protection against visible $O_3$ injury for high $O_3$ sensitive tree species such as Swiss stone pine (Sicard et al., 2016). More specific description of the algorithm and derivation of the physical relationships for the
final calculation of $POD_Y$ is given in the manual for modelling and mapping of the critical level exceedance (CLRTAP, 2015; Mills et al., 2011). The parameterization of $DO_3SE$ model reflects the recommendations in different scientific papers, the generic values are also given in manual ICP Modelling and Mapping (ICP, 2016). In this work, the preset built in version 3.0.5 of $DO_3SE$ model (SEI, 2014) with collection of parameters for coniferous forests (CF) was used (Table S1). Maximum level of stomatal $O_3$ conductance $G_{max}$ (mmol $O_3$ m$^{-2}$ PLA s$^{-1}$) of dwarf mountain pine and Swiss stone pine were obtained according
to field experiments in SK–HT. Model requires input files that include measured $O_3$ concentration and meteorological data, for each experimental site separately.



### 3.1.1 Ozone and meteorological data

In SK–HT, $O_3$ concentration was measured with active monitors (Horiba–APOA360, Thermo Electron Environmental 49C and 2B Tech Ozone Monitor M106-L) based on the well established technique of UV absorption by $O_3$ at wavelenght 254 nm.

Hourly mean data at three experimental forest stands (A, B2, C2) were recorded in continuous regime without major gaps during year 2016. Furthermore, $O_3$ concentrations measured at experimental site Lomnický štít (D) were considered for illustration of vertical $O_3$ profile in altitudinal range between 810 and 2,635 m a.s.l. In sites without electricity power (B1, C2) ozone monitors (2B Tech Ozone) were powered by solar panels. In FR–Alp (C3), passive samplers (Svenska Miljöinstitutet) were used for estimation of $O_3$ concentration. These passive $O_3$ sensors allow large-scale monitoring in remote areas (Krupa

and Legge, 2000) but do not provide real-time $O_3$ concentrations which are essential for $O_3$ flux calculations.

Method proposed by Loibl (Loibl et al., 1994; Loibl and Schmidt, 1996) was used for recalculation of aggregated data from passive samplers to hourly $O_3$ concentrations. The method is based on a model (Eq. 3) that describes daily $O_3$ profile as a function of relative elevation ($h_r$ in meters), day time hours (t from 0 to 23) and coefficients obtained from the fitting (a1, a1, a2, a3, a4 and b1, b2, b3, b4, b5, b6) (Loibl et al., 1994).

$$C(h_r, t) = a_1 + a_2 e^{-(t-a_3)^{2a_4}} \ln\left(\frac{h_r}{100} + \frac{b_1 t^2 + b_2 t + b_3}{b_4 t^2 + b_5 t + 10{,}000} e^{-b_6 t}\right) \qquad (3)$$

In FR–Alp, real-time $O_3$ concentrations continuously measured at Cians (distance about 15 km from C3) by active monitor (type Environnement SA) were used for recalculation to hourly values at C3. This approach supported small difference in seasonal average values (up to 5%). The meteorological variables (air temperature, relative humidity, wind speed, air pressure, solar radiation and precipitation) were continuously monitored at all experimental sites in SK–HT region using automatic

weather stations (Physicus, SK; EMS, CZ). In FR–Alp region, the meteorological data in hourly intervals were obtained from the station Isola 2000 (the Meteoblue weather archive at meteoblue.com) situated in the vicinity of Col de Salèse.

### 3.1.2 Stomatal conductance of montane pines

Maximum level of stomatal conductance ($G_{max}$, mmol $O_3$ $m^{-2}$ PLA $s^{-1}$) as key parameter (Eq. 1) for calculation of stomatal $O_3$ flux (Eq. 2) has not been specified neither for dwarf mountain pine nor for Swiss stone pine. Data required for $G_{max}$

parameterization were obtained by measurement on both pine species in SK–HT. For this purpose LI-6400 photosynthesis system (Li-Cor, Inc., Lincoln, NE) equipped with a standard Licor 6400-22 Opaque Conifer Chamber and 6400–18 RGB Light Source was used. To capture wide range of climatic conditions stomatal conductance was measured from June to November at study sites situated in two different elevations: site A (Stará Lesná, 810 m a.s.l.) and site C1 (Skalnaté Pleso, 1,778 m a.s.l.). During the experiment, inside chamber temperature ranged from 5 to 35 °C, VPD from 0.2 to 3.5 kPa, photosynthetic photon

flux density (PPFD) from 0 to 2,500 µmol photons $m^{-2}$ $s^{-1}$, $CO_2$ concentration was set to 400 ppm. Before each measurement, gas exchange was permitted to stabilize for approximately 6–10 min. Gas exchange measurements were conducted frequently on attached healthy looking sunlit terminal shoots of lateral branches from middle part of the plant. On average 8–9 measurements under different conditions were recorded per shoot. We consider $G_{sto}$ (mmol $O_3$ $m^{-2}$ PLA $s^{-1}$) from $G_{sto}$ (mmol





$H_2O$ m$^{-2}$ PLA s$^{-1}$) using a conversion factor of 0.663 (Massman, 1998) to account for the difference in the molecular diffusivity

of water vapour (measured by LI-6400 photosynthesis system) to that of ozone. Similar values of $G_{sto}$ with the top 5 percent in range between 110 and 160 mmol $O_3$ m$^{-2}$ PLA s$^{-1}$ were noticed for both *Pinus* species. Maximum stomatal conductance $G_{max}$ was derived as the 95 percentile of all measured data of stomatal conductance for $O_3$ flux rates (about 2,700 measurements of $G_{sto}$) after removal of outliers. Based on this derivation, maximum level of stomatal conductance for $O_3$ was determined as $G_{max}$=110 mmol $O_3$ m$^{-2}$ PLA s$^{-1}$ for both studied montane pines.

### 3.1.3 Soil water potential


Soil moisture data were obtained by two approaches: field measurement and modelling. Both, real-time and modelled data of soil water potential (SWP) are useful for specification of the $f_{SWP}$ function (Eq. 4).

$$f_{SWP} = min \left\{ 1, \left\{ f_{min} \left( (1 - f_{min}) * (SWP_{min} - SWP)/(SWP_{min} - SWP_{max}) \right) + f_{min} \right\} \right\} \qquad (4)$$

Function of $f_{SWP}$ defines the effect of soil moisture on $G_{sto}$ (Eq. 1). It is expressed in relative terms (i.e., it takes values between

0 and 1 as a proportion $G_{max}$). Additional parameters such as $f_{min}$, $SWP_{min}$, $SWP_{max}$ are listed in Table S1. Differences between $f_{SWP}$ based on measured and modelled SWP allow to verify the reliability of the soil moisture module included in DO$_3$SE model. Verification cannot be done at FR–Alp plot due to deficiency of SWP data measured in field conditions. Field measurement of SWP at three soil depths (–0.1, –0.2, –0.4 m) was carried out only in SK–HT region at all experimental sites (except the cliff of Lomnický štít). SWP values were measured with gypsum blocks in the range up to –1.5 MPa (GB2,

Delmhorst Instrument, U.S.A.) and stored with integrated data loggers (MicroLog SP3, EMS Brno, CZ). Modelling approach was used in both study areas and incorporated hydraulic resistance (steady state, SS) to water flow through the plant system (Büker et al., 2012). In model options is possible to choose appropriate alternative for simulation of soil water influence on $G_{sto}$ (Eq. 1). We selected an alternative using factor $f_{SWP}$ (Eq. 4) in calculation of $G_{sto}$. The choice with disabled $f_{SWP}$ should show different results $G_{sto}$, primarily in dry areas we expect higher values of $G_{sto}$, $F_{st}$ (Eq. 2) and consequently also POD$_y$.

### 3.2 Visible ozone injury


Observation of visible injury symptoms on Swiss stone pine and dwarf mountain pine needles collected in the autumn 2016 was undertaken in accordance with the methods recommended for analysis of the effects of air pollution on forests (ICP, 2016). Swiss stone pine branches sampled from four plots at elevation between 1,000–1,600 m a.s.l. (SK–HT) and one plot at altitude 2,000 m a.s.l. (FR–Alp) were used for visible $O_3$ injury assessment. Samples of dwarf mountain pine branches from eleven

plots situated along altitudinal profile from 800 to 2,000 m a.s.l. only for SK–HT area were evaluated. At each plot we selected 5 sample trees exposed to sun. For each tree, 5 branches with at least 30 needles per each needle age class (current year foliage (C), one year old (C+1) and two year old needles (C+2)) were removed from the upper third of the crown by using telescopic secateurs. For each branch, the percentage of total needle surface affected by visible foliar $O_3$ injury was scored for C, C+1 and C+2. Finally, a mean percentage of needles surface affected by visible foliar $O_3$ injury was calculated per every plot.



## 4 Results

### 4.1 O₃ concentration and environmental conditions

During the growing season (Apr–Sept 2016) mean $O_3$ concentrations ranged from 29.6 (B2) to 53.6 ppb (D). Measured mean hourly $O_3$ concentration (Table 2) confirmed the expected increase of $O_3$ concentration along the altitudinal profile (Fig. 2a). Therefore high altitude areas of the subalpine (C1, C3) and subnival zone (D) are most affected by $O_3$ pollution. Monthly $O_3$ means in the foothills and submontane zones varied between 20 and 40 ppb while in the subalpine and subnival zones were above 40 ppb. In SK–HT, monthly mean $O_3$ concentrations culminated in May (Fig. 2b) and achieved values between 36.3 ppb (B2) and 59 ppb (D). On the contrary, in FR–Alp (C3) the highest monthly $O_3$ average was recorded in July and reached 50 ppb. The real air temperatures (AT, deg C in Table 2) with respect to the optimum and limit range of $DO_3SE$ model parameterization (Table S1) were favourable for stomatal conductance (Fig. 3a) and potentially also for photosynthetic capacity of these two coniferous species. In FR–Alp, the average air temperature of 9.2°C at altitude close to 2,000 m a.s.l. was comparable with those of 1,600–1,800 m a.s.l. in SK–HT. Average values of vapour pressure deficit (VPD, kPa) ranged from 0.24 to 0.39 kPa (Fig. 3b) and were sufficient for unlimited stomatal conductance (Table S1). In SK–HT, precipitation totals (P, mm) in the submontane and subalpine zones varied between 669 mm and 1,280 mm and ensured a sufficient supply of soil water for roots. On contrary, low seasonal precipitation 469 mm in the foothill zone in SK–HT (A) and 425 mm in the subalpine zone in FR–Alp (C3) suggested soil drought. The amount of global solar radiation (R, kW m$^{-2}$) in FR–Alp was considerable higher than in SK–HT.

Table 2. Statistics of hourly O₃ data (ppb) and meteorological variables for the Apr–Sept 2016 period

| Exp. site | O₃ concentration (ppb) | | | | AT (°C) | | | | VPD (kPa) | | | | P (mm) | R (kW m$^{-2}$) |
| --- | --- | --- | --- | --- | --- | --- | --- | --- | --- | --- | --- | --- | --- | --- |
| | Min | Max | Mean | STD | Min | Max | Mean | STD | Min | Max | Mean | STD | Sum | Sum |
| SK–HT | | | | | | | | | | | | | | |
| A | 3.7 | 64.0 | 30.6 | 13.0 | -4.4 | 29.2 | 13.0 | 6.1 | 0.05 | 2.30 | 0.43 | 0.39 | 469 | 863 |
| B1 | 1.6 | 67.7 | 32.8 | 13.2 | -4.6 | 26.5 | 11.6 | 5.6 | 0.00 | 2.42 | 0.38 | 0.46 | 669 | 587 |
| B2 | 1.7 | 68.6 | 29.6 | 16.4 | -6.9 | 26.9 | 10.6 | 5.9 | 0.02 | 1.80 | 0.24 | 0.29 | 1,044 | 707 |
| C1 | 17.2 | 81.2 | 50.7 | 8.0 | -7.8 | 22.4 | 8.3 | 5.2 | 0.03 | 1.56 | 0.24 | 0.20 | 945 | 750 |
| C2 | 1.3 | 87.4 | 37.5 | 10.7 | -9.5 | 25.3 | 8.6 | 5.6 | 0.00 | 2.72 | 0.39 | 0.47 | 1,280 | 543 |
| D | 19.2 | 86.0 | 53.6 | 7.4 | -14.3 | 16.6 | 2.5 | 5.2 | 0.00 | 1.51 | 0.17 | 0.22 | 890 | 798 |
| FR–Alp | | | | | | | | | | | | | | |
| C3 | 17.0 | 92.1 | 48.3 | 11.3 | -6.3 | 21.6 | 9.2 | 5.2 | 0.00 | 1.81 | 0.38 | 0.27 | 425 | 931 |




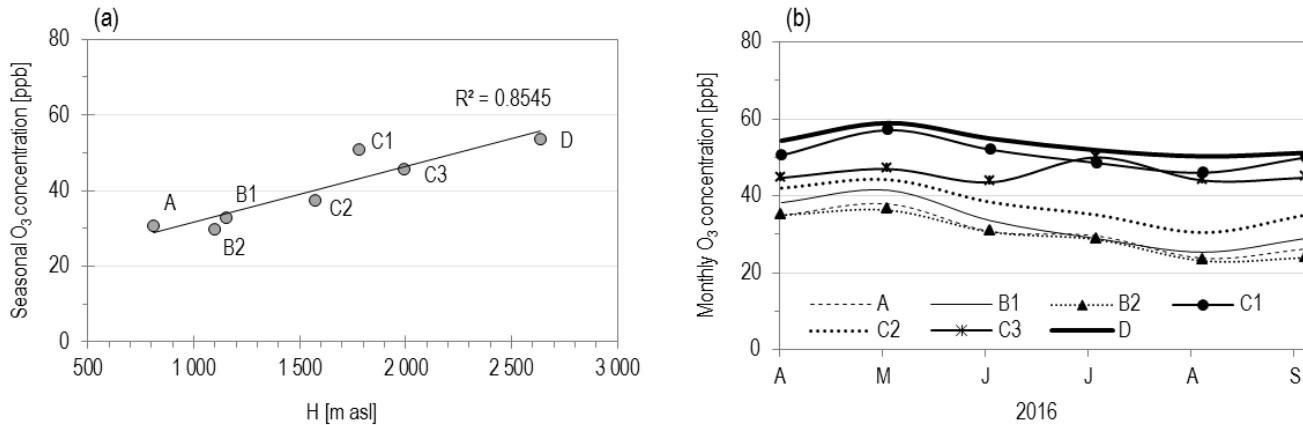

Figure 2. (a) Change of seasonal $O_3$ concentration with increasing altitude, and (b) course of monthly $O_3$ means in different altitudinal zones during the Apr–Sep 2016 period.

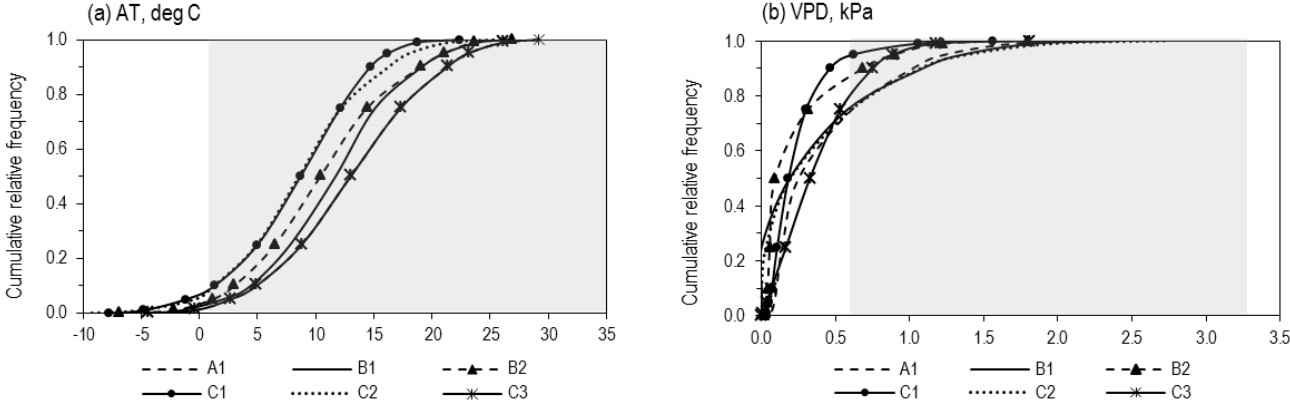

Figure 3. (a) Empirical cumulative distribution of hourly data for air temperature (AT, deg C) and (b) vapour pressure deficit (VPD, kPa) in different altitudinal zones during the Apr–Sep 2016 period; grey area illustrates ranges of AT and VPD for effective stomatal $O_3$ conductance (Table S1).

**4.2 Ozone metrics**

The DO$_3$SE model results (Table 3) show values of ozone metrics (AOT40, POD$_1$ and POD$_0$) that indicate different effect of $O_3$ pollution on mountain pines. Accumulated $O_3$ concentration exposure index AOT40 (8.3–23.3 ppm h) clearly exceeded CLec (5 ppm h) for protection of European forest (ICP, 2016) at all study sites. Phytotoxic ozone dose metric (POD$_1$) with a threshold Y=1 varied in range from 6.4 to 13.7 mmol m$^{-2}$ PLA and showed an exceedance of CLef$_1$ flux based critical level for POD$_1$ (8 mmol m$^{-2}$ PLA) proposed for coniferous forests (such as Norway spruce) at all sites in SK–HT. Accumulated stomatal $O_3$ flux without a threshold (Y=0), POD$_0$, ranged from 12.5 to 22.4 mmol m$^{-2}$ PLA for Swiss stone pine and 11.4 to 19.3 mmol m$^{-2}$ PLA for dwarf mountain pine. POD$_0$ exceeded critical level for highly $O_3$ sensitive conifers (CLef=19 mmol





m$^{-2}$ PLA) such as Swiss stone pine only at sites in SK–HT with favourable sunshine exposure (A, B2, C1). The concentration of $O_3$ and sufficient amount of soil water in submontane and alpine zones of SK–HT contributed to higher level of $POD_0$ (16.9–22.4 mmol m$^{-2}$ PLA) for Swiss stone pine compared with dwarf mountain pine (12.2–19.3 mmol m$^{-2}$ PLA). These differences between the two species may be associated with their different canopy height and root depth listed as model

parameters in Table S1. $POD_1$ and $POD_0$ values were generally higher in SK–HT than in FR–Alp. Lower stomatal $O_3$ flux and $POD_0$ value (12.5 and 11.4 mmol m$^{-2}$ PLA, for Swiss stone pine and dwarf mountains pine, respectively) which is clearly below CLef was determined in FR–Alp where the climate is considerably drier than in SK–HT. As we expected, higher $POD_0$ value (18.9 and 16.3 mmol m$^{-2}$ PLA for Swiss stone pine and dwarf mountains pine, respectively) which were closer to CLef resulted from the simulation that eliminated soil water influence on stomatal conductance (disabled soil moisture module in

$DO_3SE$ model).

Table 3. $DO_3SE$ model outputs for AOT40, $POD_1$ and $POD_0$ ozone metrics ($G_{max}$=110 mmol m$^{-2}$ PLA s$^{-1}$)

| CODE site | AOT40 (ppm h) CLec = 5ppm h | $POD_1$ (mmol m$^{-2}$ PLA) with threshold (Y=1) CLef$_1$ = 8 mmol m$^{-2}$ PLA | $POD_0$ (mmol m$^{-2}$ PLA) without threshold (Y=0) CLef = 19 mmol m$^{-2}$ PLA (highly sensitive species) | |
|---|---|---|---|---|
| | | | Swiss stone pine | Dwarf mountain pine |
| SK–HT | | | | |
| A | 8.3 | 13.3 | 21.8 | 18.4 |
| B1 | 8.9 | 10.1 | 18.1 | 15.6 |
| B2 | 13.7 | 13.7 | 21.3 | 18.9 |
| C1 | 23.3 | 14.1 | 22.4 | 19.3 |
| C2 | 13.8 | 9.3 | 16.9 | 12.2 |
| FR–Alp | | | | |
| C3 | 13.4 | 6.4 | 12.5 | 11.4 |
| C3* | 13.4 | 11.1 | 18.9 | 16.3 |

Note: *for disabled soil moisture module in $DO_3SE$ model

## 4.3 Soil moisture and stomatal conductance

The effect of soil moisture regime on $G_{sto}$ was analysed with modelled and measured SWP values and $f_{SWP}$ function (Eq. 4).

There is an acceptable agreement between SWP values modelled via $DO_3SE$ and SWP measured in SK–HT for submontane (B1, B2) and subalpine zone (C1, C2) with respect to $f_{SWP}$ (Table 4). Values of $f_{SWP}$ mostly at the level of 1 (i.e. SWP > SWP_max = –0.76 MPa) confirm the assumption that soil moisture at higher altitudes in SK–HT is sufficient for unlimited stomatal conductance and $O_3$ uptake. However, at the foothill site (A) differences between modelled and measured SWP as well as $f_{SWP}$ were larger. It appears that the model insufficiently reflected soil moisture deficit when SWP dropped below –1.2

MPa (Fig. 4). Water deficit can reduce stomatal conductance as well as stomatal $O_3$ flux to forest vegetation. The course of stomatal $O_3$ flux and $POD_Y$, taking into account the measured SWP values for foothill site (A) was tested using the algorithm recommended by the working group ICP Modelling and Mapping (ICP, 2016). In this case $POD_0$ value (20.2 mmol m$^{-2}$ PLA) was slightly lower than $DO_3SE$ model output (21.8 mmol m$^{-2}$ PLA). Validation of SWP in FR–Alp (C3) is not possible due to the absence of field measurement in 2016. In this area, the modelled average SWP (–1.02 MPa) was close to its limit (SWP_min





= −1.20 MPa). This fact, together with mean $f_{SWP}$ (0.71) suggest a decrease of $G_{sto}$ as a response to the serious soil moisture deficit in the summer season. The DO$_3$SE model using the soil moisture module application resulted in substantially decreased POD$_Y$ (C3) when the soil water influence on $G_{sto}$ in model options was considered (Table 3). Soil moisture conditions can have a significant effect on stomatal conductance (Fig. 4) and therefore for modelling of the stomatal O$_3$ flux it is important to take into account field measurement of SWP, especially in areas where soil drought events in association with lower precipitation

amount (A, C3) occur.

Table 4. Measured and modelled values of SWP and $f_{SWP}$

| Exp. site | Measurement | | | | | | DO$_3$SE model | | | | | | Difference | | | | | |
|---|---|---|---|---|---|---|---|---|---|---|---|---|---|---|---|---|---|---|
| | SWP (MPa) | | | $f_{SWP}$ | | | SWP (MPa) | | | $f_{SWP}$ | | | SWP (MPa) | | | $f_{SWP}$ | | |
| | Min | Max | Mean | Min | Max | Mean | Min | Max | Mean | Min | Max | Mean | Min | Max | Mean | Min | Max | Mean |
| SK–HT | | | | | | | | | | | | | | | | | | |
| A | **-1.44** | -0.02 | -0.46 | 0.10 | 1.00 | 0.79 | -0.08 | -0.01 | -0.02 | 1.00 | 1.00 | 1.00 | -1.36 | -0.01 | -0.44 | -0.90 | 0.00 | -0.21 |
| B1 | -0.77 | -0.02 | -0.03 | 0.98 | 1.00 | 1.00 | -0.12 | -0.02 | -0.03 | 1.00 | 1.00 | 1.00 | -0.65 | 0.00 | 0.00 | -0.02 | 0.00 | 0.00 |
| B2 | -0.02 | -0.01 | -0.02 | 1.00 | 1.00 | 1.00 | -0.04 | -0.02 | -0.02 | 1.00 | 1.00 | 1.00 | 0.02 | 0.01 | 0.00 | 0.00 | 0.00 | 0.00 |
| C1 | -0.05 | -0.03 | -0.04 | 1.00 | 1.00 | 1.00 | -0.04 | -0.02 | -0.02 | 1.00 | 1.00 | 1.00 | -0.01 | -0.01 | -0.02 | 0.00 | 0.00 | 0.00 |
| C2 | -0.03 | -0.02 | -0.03 | 1.00 | 1.00 | 1.00 | -0.02 | -0.01 | -0.01 | 1.00 | 1.00 | 1.00 | -0.01 | -0.01 | -0.02 | 0.00 | 0.00 | 0.00 |
| FR–Alp | | | | | | | | | | | | | | | | | | |
| C3 | | : | | | : | | **-4.00** | -0.02 | -1.02 | 0.10 | 1.00 | 0.71 | | : | | | : | |

Note: Values in bold are SWP < SWPmin = −1.20 MPa for CF (Table S1)

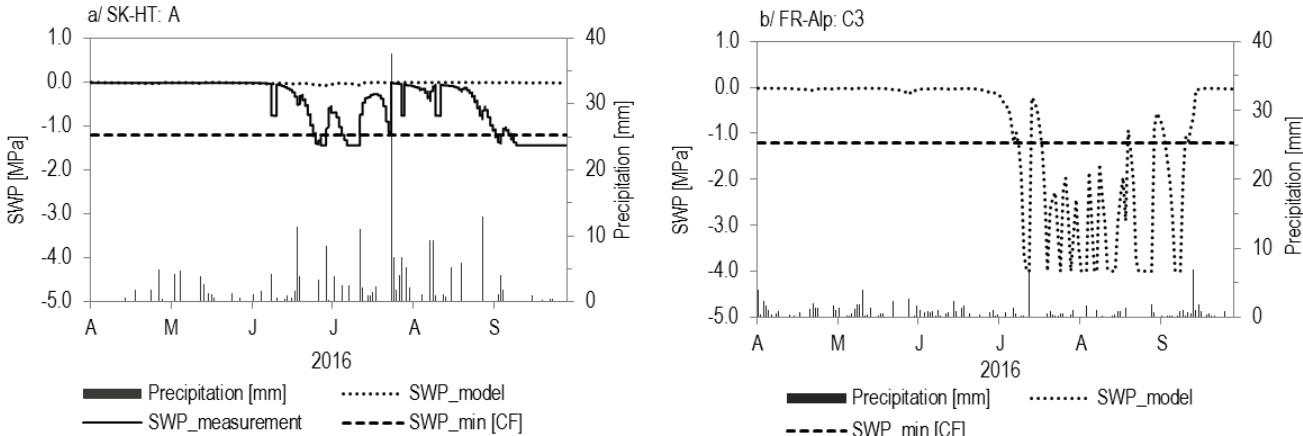

Figure 4. Hourly data of precipitation (mm) and soil water potential (SWP, MPa) for two experimental sites: (a) A in SK–HT, and (b) C3 in FR–Alp as response to dry events which occurred during summer 2016. SWP$_{min}$ (−1.2 MPa) is a limit value of
soil water potential for minimal stomatal conductance for the coniferous forest (CF) (Table S1).

**4.4 Visible ozone injury**

Swiss stone pine and dwarf mountain pine branches sampled along the vertical profile from 800 to 2,000 m a.s.l. showed an obvious visible O$_3$ injury at higher altitudes for both bioclimatic regions (Fig. 5). In SK–HT plots, more pronounced visual symptoms were observed for dwarf mountain pine (7.7 ± 1.1 % for C+1 needles and 18.2 ± 2.3 % for C+2 needles) than for



Swiss stone pine (2.2 ± 0.4 % for C+1 needles and 7.2 ± 2.0 % for C+2 needles) plots. The oldest needles of both species were more frequently damaged by O$_3$ injury at all plots. The youngest, current year needles did not show any signs of O$_3$ injury. No significant differences were found between southern and northern transects. Higher damage of dwarf mountain pine by O$_3$ could be caused by mild winter of 2015/2016 with unusually low snow cover in SK–HT plots. The dwarf mountain pine which is usually covered by snow in spring was exposed to ambient air O$_3$ maxima in 2016.

Chlorotic mottle and marbling as characteristic markers of O$_3$ damages were clearly visible on the Swiss stone pine in FR–Alp plot with the highest percentage of visible O$_3$ injury from 10.0 ± 0.6 % for C+1 needles and 25.0 ± 0.9 % for C+2 needles, respectively. This observation confirms that the Swiss stone pine could be considered as a sensitive bioindicator of O$_3$ exposure in the FR–Alp area.

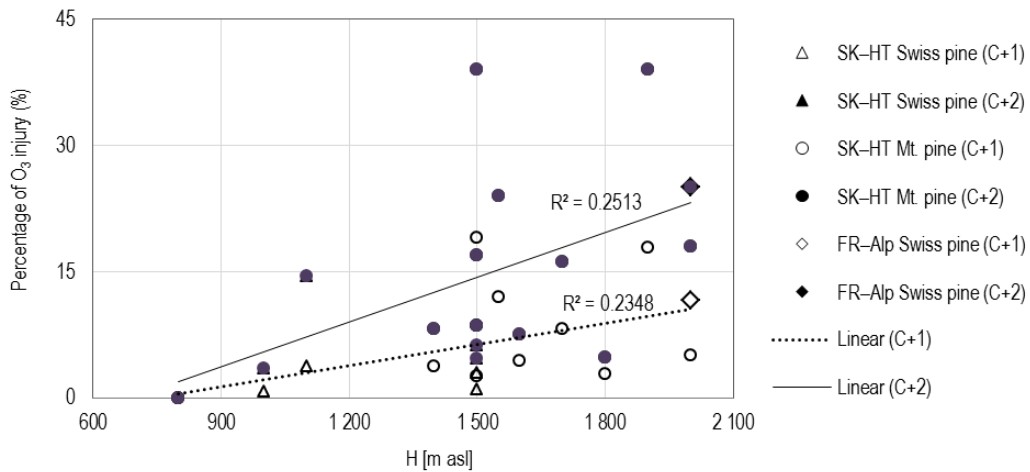

Figure 5. Percentage of O$_3$ injury visible on needles of different age for pines within the SK–HT and FR–Alp regions.

## 5 Discussion

Estimation of phytotoxic effect of ozone on coniferous trees at high elevation and vulnerable mountain forests needs a special approach. Model simulation of stomatal O$_3$ uptake requires continuous field measurement of hourly O$_3$ concentration and various meteorological parameters. Precise parameterization of the DO$_3$SE model is also important for calculation of an

accumulated stomatal O$_3$ flux i.e., POD$_Y$ metrics. Such data is widely available for some commonly occurring trees species, such as Norway spruce (ICP, 2016; Mills et al., 2011) with low sensitivity to O$_3$ exposures (Coulston et al., 2003). However, there is still a lack of empirical data concerning potentially vulnerable mountain forest tree species. Therefore modification of input parameters reflecting real environment of specific coniferous species may be very useful, especially maximal stomatal conductance of mountain pines in different bioclimatic regions.

In this work, we modified maximal stomatal conductance G$_{max}$=160 mmol O$_3$ m$^{-2}$ PLA s$^{-1}$ generally used in DO$_3$SE model as a standard for coniferous tree species. Based on real-time measurements of mountain pines in SK–HT we changed this preset value to the adjusted G$_{max}$=110 mmol O$_3$ m$^{-2}$ PLA s$^{-1}$ (Table S1). This modification resulted in POD$_0$ values substantially lower



(about 10 mmol m$^{-2}$ PLA) than those predicted by the DO$_3$SE model with preset G$_{max}$ value. Further change of POD$_0$ value can be expected after modification of G$_{max}$ based on real-time measurements of G$_{sto}$ in much warmer FR–Alp conditions.

Our modelling of POD$_0$ showed that FR–Alp experienced lower O$_3$ uptake than the sites in SK–HT. Contrary to this, visible O$_3$ injury of Swiss stone pine was more serious in FR–Alp than in SK–HT. This might be influenced partly by different altitudes of Swiss stone pine plots where visible O$_3$ injury observations were carried out: in FR–Alp at elevation of 2,000 m a.s.l., and in SK–HT at elevations between 1,000 and 1,600 m.a.s.l.. In addition, it is necessary to consider some degree of uncertainty of the DO$_3$SE model in animation of stomata functioning driven mostly by soil water potential. Real low soil

moisture in FR–Alp should notably reduce stomatal conductance, and consequently yield low values of POD$_0$ leading to avoidance of visual O$_3$ symptoms. Based on a strong visual evidence of injuries we suspect that the used model underestimated stomatal conductance. The importance of correct incorporation of soil moisture into modelling of stomatal O$_3$ uptake was recently reported by De Marco et al. (2016).

Sufficient amount of precipitation and favourable soil moisture eliminated influence of f$_{SWP}$ (Eq. 4) on stomatal conductance

(Eq. 1) at most of the study sites in SK–HT. This fact caused that f$_{SWP}$ was nonessential for the POD$_Y$ calculation (Eq. 2). On the contrary, in FR–Alp the modelled SWP values were markedly lower and the POD$_0$ values changed. These findings underline the necessity to include site specific real–time SWP data into the DO$_3$SE model.

Beside these findings the study revealed an importance of the nighttime plant transpiration and stomatal conductance for more precise estimation of POD$_o$ values. The DO$_3$SE model sets the night time values to zero and thus understimate stomatal

conductance during night. Our preliminary night time physiological measurement on dwarf mountain pine (unpublished data) indicate the role of night time fluxes similarly as in other tree species (Zeppel et al. 2013).

An earlier onset of real growing season and its extended duration in the FR–Alp region could be another explanation of discrepancy between serious visual injury symptoms and relatively low POD$_0$ value. For the purpose of comparison and limited data availability only the period from April to September 2016 was considered. It is possible that relevant O$_3$ uptake was

present in time prior to the considered period. All round year field measurement campaign at higher altitudes of Mediterranean mountains should be realized to verify this assumption.

Based on the outputs of our study we can confirm that visible O$_3$ injury increases with rising altitude (Díaz-de-Quijano et al., 2009; Benham et al., 2010; Kefauver et al., 2014). A fact that O$_3$ concentrations and O$_3$ uptake into the foliar tissues rise with altitude is supported by many previous studies (Chevalier et al., 2004; Bičárová et al., 2013). Our findings concerning visible

O$_3$ injury and altitude confirm findings from other studies (e.g. Dalstein et al., 2004; Ulrich et al., 2006).

Our observations of visible symptoms indicate that, the Swiss stone pine in the Carpathians seems to be less sensitive to O$_3$ and consequently show lower visible injury than dwarf mountain pine. Although Mediterranean vegetation is considered to be better adapted to oxidative stressors than mesophilic vegetation (Paoletti, 2006), we found that in FR–Alp the visible symptoms on Swiss stone pine (chlorotic mottles and marbling) were more evident, evolved over time and their number increased with

age. The marbling was also found on the oldest needles and a cumulative effect was more pronounced. Different visible O$_3$ injury response may be expected under natural conditions due to differences in O$_3$ sensitivity controlled by a genotype and



micro-site conditions of growth, exposure, and $O_3$ flux (Coulston et al., 2003; Nunn et al. 2007; Braun et al., 2014). Despite genus mutuality of Swiss stone pine and dwarf mountain pine we found surprisingly notable differences in their visible $O_3$ injury. It raised questions if this is the result of different species tolerance to $O_3$ injury and different rate of $O_3$ uptake or simply an inadequacy of the visual injury assessments. Evaluation of $O_3$ injury symptoms on conifers is not easy and therefore may produce some questionable results as discussed by several authors (e.g. Wieser et al., 2006). Although visible injury is commonly used as an indicator of phytotoxic $O_3$ concentrations in ambient air, it is not always a reliable indicator of damage or other injury endpoints (EPA, 2007). Due to high sensitivity of Swiss stone pine proven in FR–Alp it can be considered as local bioindicator of $O_3$ exposure. In SK–HT, dwarf mountain pine with pronounced visual symptoms seems to be an appropriate local conifer species for future biomonitoring of $O_3$ injury.

## 6 Conclusions

Swiss stone pine and dwarf mountain pine are typical tree species of the European timberline. Determinations of selected $O_3$ phytotoxicity metrics (AOT40, $POD_1$, $POD_0$) with respect to exceedance of relevant critical levels (CLec, $CLef_1$, CLef) suggests different adverse $O_3$ effect on mountain pines in two contrast bioclimatic regions: moist in the High Tatra Mts in Slovakia (SK–HT) and dry in the Alpine Mercantour Mts in France (FR–Alp). Values of AOT40 were found to be substantially higher than the critical level CLec for both moist and dry regions. However, modelled results of $POD_0$ values that reflect influence of climatic and environmental conditions on the opening of the stomata showed lower $O_3$ uptake to conifer needles in the Mediterranean region of the Alps, where the climate is considerably drier than in the Carpathians. Humid environment leads to a greater capture of $O_3$ gas in tissues of plants exposed to high $O_3$ doses. Observations of visible $O_3$ injury confirmed that older needles were more damaged by $O_3$ than the younger ones. Despite the favourable humid soil conditions for stomatal $O_3$ uptake, we observed relatively weak visible $O_3$ injury on two year needles of Swiss stone pine at SK–HT. Dwarf mountain pine with more pronounced visual symptoms seems to be an appropriate conifer species for further monitoring of $O_3$ injury at SK–HT. Severe foliar $O_3$ injury symptoms were identified on Swiss stone pine at the FR–Alp experimental site.

Presented results confirm high phytotoxic potential of $O_3$ air pollution in different bioclimatic regions of Europe, although the biologically-based field data do not completely correspond with the calculated $O_3$ metrics. For better explanation of differences in visual $O_3$ symptoms, further studies concerning $O_3$ resistance of pine species in changing real soil humidity regime in subalpine zone are needed. In future research, closer attention should be paid to the entire growing season as well as night time $O_3$ fluxes, since both these time spans appear to play an important role in estimates of POD in mountain forests.

## Team list

**Author contribution**: S. Bičárová, H. Pavlendová, Z. Sitková designed and carried field measurement of $O_3$, meteorological variables and SWP in SK–HT; S. Bičárová calculated $O_3$ metrics; H. Pavlendová assessed visual $O_3$ injury and Z. Sitková analysed SWP data in SK–HT; P. Fleischer jr. and P. Fleischer sr. designed and carried field measurement of stomatal conductance for dwarf mountain pine and Swiss stone pine; L. Dalstein-Richier and M. L. Ciriani prepared data for FR–Alp experimental site; A. Bytnerowicz suggested methodological improvements of field experiments and helped with preparing and writing of the manuscript. All authors discussed the results and contributed to the final manuscript.





**Competing interests:** The authors declare that they have no conflict of interest.

**Supplement** Part SI–1: Details in section Methods: DO$_3$SE model parameterization (Table S1)

**Acknowledgements**
This work was supported by the Slovak Research and Development Agency under the contracts No. APVV–0429–12, APVV–16–
0325, and by the Grant Agency of the Slovak Republic (VEGA, No.2/0053/14 and No. 2/0026/16). We acknowledge also the project
ITMS 26220220066 funded by ERDF (10 %). The authors are grateful to the Slovak Hydrometeorological Institute (SHMI)
for providing of meteorological, climatic and EMEP data. The development of DO$_3$SE model interface has been made
possible through funding provided by the UK Department of Environment, Food and Rural Affairs (Defra) and through
institutional support provided to the Stockholm Environment Institute from the Swedish International Development Agency
(Sida).

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
