# Peer review of "How does soil water availability control phytotoxic O₃ dose to montane pines? Modelling and experimental study from two contrasting climatic regions in Europe"

_Atmospheric Chemistry and Physics, 2017_

## Referee Comment (RC1) · Anonymous Referee #1 · 20 Nov 2017

The results organization, the quality of interpretations and conclusions must be improved. Some parts are particularly confused and results cannot be easily interpreted. On several points it is unclear how the conclusions have been obtained. The inclusion of the French site includes a lot of inconsistencies and no added-value to the paper. I suggest submitting this paper under a journal with lower Impact Factor, removing the French site and updating the list of co-authors if need be.

++ Major opened issues need to be clarified.

PODy must be calculated from on-site data, not from data coming from distant stations. This issue is critical. Indeed, the ozone concentrations are measured at more than 15km from site FR-Alp and meteorological data from another station (> 20 km). The ozone pattern and meteorological parameters are heterogeneous, in particular in mountains. The SWP measurements are missing for this site. Which parameterization did you use, i.e. Continental Central European species for coniferous forests? Please clarify this issue.

Line 160: the maximum stomatal conductance (gmax) is based on the average above the 95th percentile of gsto measurements under optimum environmental conditions for stomatal opening. The stomatal conductance measurements ranged from 110 to 160 mmol O3 m-2 PLA s-1, how is possible to have a 95-pencentile of 110? Norway spruce median value 125 mmol O3 m-2 PLA s-1, range (87-140), is used for mature Norway spruce trees (CLRTAP, 2015) while you have specified a value of 160 mmol O3 m-2 PLA s-1 generally used in DO3SE model as a standard for coniferous tree species. This is false.

What is the validity of ozone-induced injury data? Who carried out the epidemiological studies? Trained experts (e.g. ICP-Forest experts)? Authors have to stress on the in-field data quality and on the importance to validate these data by experts for a better performance. How did you consider the data variations between surveyors? Did you find dwarf mountain pines in Mercantour park (table 3)?

The POD0 for FR-Alp (12.5 mmol O3 m-2) seems too low. Did you include SWP in PODy calculation for the French site? Line 260: did you test (statistically) the difference? On Figure 5, the both values FR-Alp for C+1 and C+2 appears as outliers. The large difference for the percentage of surface affected by ozone-induced injury between both regions can be due to the difference of field evaluation by surveyors. This point could be discussed.

As you have calculated PODy, differently (e.g. SWP inclusion), the comparison between both regions is inconsistent. The authors observed a lower percentage of visible ozone injury while higher POD0 were calculated. This conclusion must be carefully argued. Indeed, PODy must be calculated over the accumulation period started from the start date of the growing season until the day when the survey of symptoms was carried out. This was not respected, therefore PODy is overestimated. In addition, the ozone and meteorological data are from distant stations in France and SWP was not measured in France.

Line 305: SWP non-essential for the PODy calculation? Please explain this contradictory point.

Line 325: why did you mention "Different visible O3 injury response may be expected under natural conditions..."? You are under real world conditions.

++ A series of grammatical and typos errors, throughout the manuscript, must be taken into account.

The title is too long; the second part of the title can be removed. Y represents a detoxification threshold, rather than "threshold flux" (line 43-44), below which it is assumed that any ozone molecule absorbed by the plant will be detoxified. Line 47: what are the reasons to use POD0 rather than POD1?

Line 104: AOT40 is calculated for hours with a short wave radiation exceeding 50 W m-2 according to CLRTAP (2015), commonly the time window 8am-8pm is used at 45° latitude. The critical value of 5 000 ppb h was established for forest protection (line 105).

Line 106-114: gsto (mmol O3 m-2 PLA s-1) is estimated from functions describing the response of stomata to key species-specific and environmental variables. Please include the formula PODy, define fmin, what are the values rc and rb? What is the integrated time-window? The functions, not defined in this paper, are expressed in relative terms (between 0 and 1). Define the function fo3 correctly.

In the Mediterranean area, the functions of SWC and phenology are considered redundant. Why did you assume that fphen was 1 throughout the growing season, i.e. the PODY accumulation period? Line 116-121: move to section "Discussion". Where is the photosynthetically active radiation?

For the SWP, did you follow the "Part X: Sampling and Analysis of Soil" protocol (ICP-Forests manual) for field campaigns to measure Field Capacity and Wilting Point?

Line 134: what is the integration time for passive samplers (weekly, bi-weekly)?

Line 293: the difference of 10 mmol O3 m-2 PLA seems high.

Line 295: why the ozone uptake is lower? Line 308-309: this finding is not based on the current paper, please add references.

Where are values of PODy calculated from passive samplers? The method based on Loibl formula to describe the daily profile of hourly ozone concentrations has one major limitation, i.e. not suitable for high POD thresholds, and that accuracy of the measurements with passive samplers has to be strictly assured in order to finally obtain acceptable errors (Calatayud et al., 2016).

The difference for PODy, with and without SWC, increases with increasing Y (De Marco et al., 2016). The differences were lower for temperate species (Pinus cembra, Abies alba) than for Mediterranean species. SWC must be considered in PODy simulations and a low Y threshold should be used for robustness.

Missing references (e.g. line 90, section 3.2).

Figure 1 must be improved (North arrow, scale...). Table 2: please define acronym in caption. Why did you sum the global radiation? The R values seem too high at 45° latitude, please check. Table 4: for the site C1 the error is of 50%, not "acceptable" as specified within the manuscript (line 236). Figure 2: please name X-axis.

---

## Referee Comment (RC2) · Anonymous Referee #2 · 18 Dec 2017

While providing some interesting insight into the ozone pollution climate at two mountainous sites across Europe and the respective ozone uptake of two tree species and their associated physiological characteristics , unfortunately this study is flawed with respect to its methodology and general focus and therefore not publishable in the current form. A major revision is necessary, after which the study could be submitted to another, lower impact journal. The high scientific standards of ACP – especially with respect to relevance and novelty - cannot be met with this study. The reasoning behind this rejection is listed in the following.

Introduction not very comprehensive (effects of ozone on tree physiology could be better described) and in parts imprecise (for example discussion about choice of y-threshold); it lacks some key information, e.g. purpose of risk assessments? Why are you interested in a comparison between AOT40 and POD (because you don't discuss the difference between different ozone metrics, readers won't see the point of introducing different metrics)? Why are you interested in doing a precise risk assessment of ozone effects on the two pine species, when you don't show flux-response (visible injury) relationships in the end? Who will be interested in your study, and why? Also, what are the known problems with response parameter "visible injury" as compared to growth increment? And the introduction doesn't really explain why you concentrated on those two mountain regions and those two tree species. In fact, the benefit of including the French site in the study is not clear, also because of the questionable quality of the ozone data as compared to the Slovakian site.

The "study area" chapter needs a revision: What is the link between the several Slovakian and one French site? Why only one site in France but several in Slovakia? The combination of the these two provenances is not clear (Pinus cembra only occurs in one Slovakian site, and only sparsely; Pinus mugo does not occur in the French site at all) and sounds very "random";

The methods chapter needs a thorough revision as it is fundamentally flawed in its current form! The DO3SE parameterisation is key to the study and the respective table should be provided in main text; is the parameterisation really the same for both Pinus species at both sites (apart from canopy height and root depth)? This is very questionable and needs some proper scientific discussion and proof based on measurements. The main parameters I would question in this respect are the gmax and fphen – I would be very surprised if they were the same at both locations and for both species. For this, you will have to show (in a figure) and statistically analyse the physiological (gsto) measurements you did – which raises the problem that they were only done in Slovakia and not in France. Without respective measurements in France, this study

can't be published. Why don't you show flux-response (visible injury) relationships in the paper? This would have been the natural final result readers might expect from this study. Also, the choice of threshold (Y = 0 vs. Y = 1) has to be statistically explained. This is chapter lacks key references to DO3SE too.

Ozone and meteorological data: You describe how you derive hourly data from measurements done with passive samplers at the French site, but did you also actually cross-calibrate the passive samplers with the active samplers before exposure? This needs to be explained. Also, how many measurements were made with the passive samplers? When were the passive samplers replaced, how long was their exposure time? And how far is the meteorological site in France away from the ozone sampling site? The C1 site is not referred to in this chapter, but C2 is mentioned twice. A typo?

Soil water potential: Another problem with the French site, as SWP was not measured there. Which approach did you use for the DO3SE modelling at that site (you don't explicitly describe this) and how did you validate the modelled SWP? This is another strong argument for excluding the French site altogether.

Results: Figure 3b seems to be wrong: The uptake of ozone starts to be limited above a certain VPD, here 0.6 kPa. So the grey area shown in that figure shows the non-effective uptake, which is in exact contrary to what you write. Flux- and concentration (why do you otherwise include AOT40 in your study?) response relationships using the visible injury data would have been expected at the end of this study/paper.

Discussion: In parts very thin, specifically with respect to the influence of SWP on ozone flux given the suggested SWP focus provided in title of paper! There are a multitude of key publications in this field the study results should be compared to. Also, no discussion of AOT40 vs. POD, which presumably is partly based on the omission of showing dose-response relationships.

Please do a general spelling and language test – there are typos, punctuation and grammatical errors and in general the writing is sub-standard, often not precise enough

Some specific comments: L 36-37: "based only on measured O3 concentration does not take into account environmental factors affecting responses of vegetation" – what kind of response? Not precise enough, a frequent problem in the manuscript L. 41: add more recent DO3SE literature, such as Emberson et al., 2007, Büker et al, 2012, L. 45: There is an updated version of Karlsson et al. 2007 available that should be cited in addition: Büker et al, 2015 L 55: "The most sensitive conifers are Pinus species" – sensitive in terms of what? Foliar injury? There are a lot of broad-leaf tree species that are very sensitive too – please be more precise L 55/56: "however different visible O3 injury response may be expected under natural conditions" –What do you mean exactly? L 62: Please rephrase objective 1: What exactly do you mean by assessing metrics? L 90: "Visible leaf injury on particularly sensitive species is one of the O3 air pollution symptoms" – please give reference Table 1: There is no tree vegetation at site D? Why then include it, as this is a paper focussing on forest trees? L 103: DO3SE model needs references mentioned in this chapter, even if they have been listed before L 105: What is the "relevant" growing season? This is not specified for the French site earlier; do both locations really have the same growing season? L. 108: "Passage rate" – please refer, this is not a scientific term L 117/120: Please check latest updates on CLs on ICP Vegetation website L 117: "An innovative species-specific…" - this sentence needs much more detail; for example, explain why a POD0 rather than POD1 was used L 247: "Soil moisture can have.." – this is content that should be included in discussion, not results section.

---

## Referee Comment (RC3) · Anonymous Referee #3 · 19 Dec 2017

I wouldn't recommend the publication of the manuscript entitled "How does soil water availability control phytotoxic O3 dose to montane pines? Modelling and experimental study from two contrasting climatic regions in Europe" for several scientific weakness. First of all the title is misleading. Indeed I expected a discussion about the role of soil water content in determining the POD in montane pine, while this aspect is not developed into the manuscript. The measurements of ozone that are the basis for POD calculation are obtained by passive samplers and is well known by the ozone community that these measures are integrated over a period variable in time window and that

are not appropriate to obtain hourly data. Then the authors stated that soil moisture data are not measured in French sites, thus is not clear the importance to include into the manuscript the French site, even considering that is a single one, in comparing with higher number of sites in Slovakia. Another critical point is that the authors declare that the meteorological measurements are available in France not in the same site of ozone measurements but far from there, without specify if the climate of the meteorological station is comparable with the ozone station. About the methodology is not clear how the authors consider the function fphen included in DO3SE model. Did they consider it as a constant in the France and Slovakia sites? This is meaningless because of the stric t link between phenology and climatic conditions. Another weakness of the manuscript is that the author described two different indicators to explain the visible injuries occurrence due to ozone pollution, but they didn't show dose-response relationship between the two indicators and the symptom's occurrence. This is an important point to describe and discuss into the manuscript, or the other option is to limit the manuscript to a descriptive observation of two indicator of potential ozone damages to forests trees. Thus I suggest to remove the part related to ozone injuries or to include dose-response relationships.

---

## Author Comment (AC1) · 5 Jan 2018

Subject: Authors' Response to all Referee reviews (Referee #1, Referee #2, Referee #3)

Dear Referees,

We really appreciate your effort and valuable although critical comments regarding to manuscript No. acp-2017-1005. We accept your statement for major revision of manuscript as well as recommendation to submit of revised paper to another type of journal. Currently we work on completely new version of manuscript that will consider most of your comments. Please find enclosed supplement including our reply to principal problems.

On behalf of all co-authors, yours sincerely, Svetlana Bičárová

In the following, answers to principal problems are shortly described.

**Problem:** Improving of the results organization, interpretation and conclusions.
**Answer:** We accept this comment. Revised manuscript will focus on the role of environmental factors in process of $O_3$ uptake only to dwarf mountain pine in Slovak study site in the High Tatra Mts. (SK–HT). We also accept comment concerning English Language revision. As English is not our native language, upon completion of the professional discussion, the text will be sent to a professional linguistic correction.

**Problem:** The inclusion of the French study site and on-site data in FR-Alp plot (C3)
**Answer:** Modelling of stomatal ozone flux requires complex inputs based on real measurements and specific parameters. Database of measured input variables in FR-Alp is not supported by measurement system focused on modelling of stomatal ozone flux. We worked with the available data that were modified. Of course, we agree that the best is to have data from one location. At one study site in FR–Alp (plot C3, Col de Salèse), the ozone symptoms on Swiss stone pine were assessed and there were also measured ozone concentrations using passive samplers. The following data sources were used to prepare the input file for the PODy calculation.

Ozone data
- real measurement (passive samplers) on plot C3-Col de Salèse with seasonal $O_3$ mean of 46 ppb
- real measurement (active ozone analyzer) on site Cians with relatively similar seasonal $O_3$ mean of 43 ppb
Diurnal variability of hourly data from the Cians ozone analyzer were modified according to eq. 3 (in manuscript) and then recalculated with respect to monthly mean differences. Partial steps of this modification illustrate figures below this answer.

Meteorological data
- hourly data from the nearest Isola 2000 meteorological station were used in the input file
According to the average air temperature for the period from April to September, the climate at C3-Col de Salèse (13.8 ° C) is similar to that of Isola 2000 (13.7 C).

Although collection of one-site real hourly $O_3$ and meteorological data at C3 plot is serious problem, modification of available data provides rational framework for model estimation of PODy. Nevertheless, revised manuscript will not include French study site. In the future, increase attention should be paid to extension of real field measurements of ozone, meteorological and environmental variables on timberline zone of mountains in Europe.

[Figure]

**Problem**: Parameterization for coniferous forest (CF)

**Answer:** We used parameterization for coniferous forest according to built-in preset in model DO3SE (see below). List of complete parameters is included in supplement Part SI-1: Details to section: Methods DO3SE model parameterization for version (DO3SE_INTv3.0.5)..

[Figure]

**Problem:** Maximum stomatal conductance Gmax for Swis pine is 110 mmolO$_3$ m$^{-2}$ PLA s$^{-1}$. Is it true or false?

**Answer:** Next box plots illustrate Quantiles estimation of measured values of stomatal conductance Gsto SK–HT region. In this study we defined 95-Percentil as maximal stomatal conductance Gmax. After rounding it is value of 110 mmolO$_3$ m$^{-2}$ PLA s$^{-1}$ for both Swiss pine and dwarf pine. Median or 50-Percentile values between 50 and 60 mmolO$_3$ m$^{-2}$ PLA s$^{-1}$ are substantially lower and, do not correspond at all with median value of 125 mmolO$_3$ m$^{-2}$ PLA s$^{-1}$ for Norway spruce (Continental Central Europe). Norway spruce median value 125 mmolO$_3$ m$^{-2}$ PLA s$^{-1}$ referred e.g. in Körner et al. (1979), Dixon et al. (1995), Emberson et al. (2000), Zweifel et al. (2000, 2001, 2002) was derived from range of values between 87 and 140 mmolO$_3$ m$^{-2}$ PLA s$^{-1}$. This range is probably related to variability of Gmax values.

[Figure]

**Problem:** The validity of ozone-induced injury data

**Answer:** The visible ozone symptoms assessment was carried out by the national experts of ICP Forests, Expert Panel on Ambient Air Quality, who was trained at intercalibration courses on visible ozone symptoms. Variation of surveyors was not assessed. Other types of international training workshops would be useful.

**Problem**: For the SWP, did you follow the Part X: Sampling and Analysis of Soil protocol (ICP Forests manual) for field campaigns to measure Field Capacity and Wilting Point?

**Answer:** Yes, we partially followed the methodology of ICP Forest Manual (part X), as the Pedological characterisation and detailed soil profile description at our plots was complemented by sampling according to genetic horizons and a detailed soil classification was based on the World Reference Base for Soil Resources (IUSS Working Group WRB, 2015). Final soil type of each site was displayed in Table 1 in manuscript. Besides, at all research localities we continuously monitored soil water (matrix) potential (SWP, MPa) at three fixed depths and in three different soil profiles to catch hydropedological variability of each site (methodology fully in line with ICP Forests Manual, part IX Meteorological measurements). Detail analyses and laboratory determining of pF retention curves (wilting point and field capacity) were not the objectives of this study. But manufacturer of sensors for SWP measurements (Gypsum blocks, GB-2 Delmhorst Instrument, Co. and MicroLog SP3, EMS Brno, CZ) declare the limit value of -1,5 MPa as wilting point, when the soil water becomes unavailable for forest trees.

---

## Author Comment (AC2) · 5 Feb 2018

Dear Referee #1, We really appreciate your effort and valuable although critical comments regarding to manuscript No. acp-2017-1005. We accept your statement for major revision as well as recommendation to submit of revised paper to another type of journal. We have prepared completely new version of manuscript that considered most of your comments. Our answers are enclosed in supplement titled as "Authors' Responses to Referee #1 Comments.pdf".

[Figure]

On behalf of all co-authors, yours faithfully, Svetlana Bičárová

Please also note the supplement to this comment:
https://www.atmos-chem-phys-discuss.net/acp-2017-1005/acp-2017-1005-AC2-supplement.zip
* * *

---

## Author Comment (AC3) · 5 Feb 2018

Dear Referee #2,

We really appreciate your effort and valuable although critical comments regarding to manuscript No. acp-2017-1005. We have prepared completely new version of manuscript that considered most of your comments. Please find enclosed supplement including our reply to principal problems as well as to your specific comments.

On behalf of all co-authors, yours faithfully, Svetlana Bičárová

[Figure]

Please also note the supplement to this comment:
https://www.atmos-chem-phys-discuss.net/acp-2017-1005/acp-2017-1005-AC3-supplement.zip
* * *